# A Study of Common Principles for Decision-Making in Moral Dilemmas for Autonomous Vehicles

**DOI:** 10.3390/bs12090344

**Published:** 2022-09-19

**Authors:** Li Li, Junyou Zhang, Shufeng Wang, Qian Zhou

**Affiliations:** Faculty of Transportation, Shandong University of Science and Technology, Qingdao 266590, China

**Keywords:** automated driving, moral dilemmas, behavioral decisions, common principles

## Abstract

How decisions are made when autonomous vehicles (AVs) are faced with moral dilemmas is still a challenge. For this problem, this paper proposed the concept of common principles, which were drawn from the general public choice and could be generally accepted by society. This study established five moral dilemma scenarios with variables including the number of sacrifices, passenger status, presence of children, decision-making power subjects, and laws. Based on existing questionnaire data, we used gray correlation analysis to analyze the influence of the individual and psychological factors of participants in decision-making. Then, an independent sample *t*-test and analysis of covariance were selected to analyze the influence relationship between individual and psychological factors. Finally, by induction statistics of decision choices and related parameters of participants, we obtain common principles of autonomous vehicles, including the principle of protecting law-abiding people, the principle of protecting the majority, and the principle of protecting children. The principles have different priorities in different scenarios and can meet the complex changes in moral dilemmas. This study can alleviate the contradiction between utilitarianism and deontology, the conflict between public needs and individualized needs, and it can provide a code of conduct for ethical decision-making in future autonomous vehicles.

## 1. Introduction

With the rapid development of artificial intelligence, autonomous vehicles (AVs) are an inevitable trend for future transportation development. They have the advantages of reducing traffic accidents, improving traffic efficiency, reducing fuel costs, and protecting the environment [1,2]. Under unexpected conditions such as the failure of the sensory equipment of AVs and the sudden entry of objects in front of them [3], AVs will encounter situations where collision is unavoidable, i.e., a moral dilemma.

Although these emergencies (moral dilemmas) occur in a split second, we still need to study such low probability accidents in order to protect human lives. At the same time, studies have shown that people may refuse to buy AVs if there is no clear moral algorithm to guide the decision-making of autonomous driving systems [2]. Therefore, how to make moral decisions that are acceptable to the public is an important challenge for AVs.

Current research on decision-making in response to moral dilemmas has focused on the conflict between choosing utilitarian or deontological moral principles, and the contradiction between public and individual needs. But few studies have been conducted on the common principles that can be universally followed by AVs. To enhance the trust and confidence of consumers in AVs, this study adopts the moral dilemma of AVs as the research context based on existing theoretical approaches. The common principles of moral decision-making that can be universally accepted by society and the public are investigated. Moreover, the principle can be used as a premise and basis for moral dilemma decision-making in AVs.

This paper is organized as follows: Section 2 contains a review of the current literature related to moral dilemma decision-making in AVs. Section 3 presents the moral dilemma scenarios and the corresponding questionnaire data, and also describes the data analysis methods which were used. Section 4 contains the results of Statistical Product and Service Solutions (SPSS) independence analysis and gray relational analysis to obtain the effects of age, gender, and psychological factors on decision-making under the role of variable factors in each scenario. Section 5 discusses the information extracted from different scenarios and summarizes the common choice characteristics of members of the public in AVs moral dilemma decision-making. Finally, in Section 6, we derive the order of the principles of protection of the majority, protection of children, and protection of the law under different conditions.

## 2. Literature Review

This section focused on research related to three aspects of AVs: moral dilemma, deontological versus utilitarian moral preferences, and public versus individualized needs.

### 2.1. Moral Dilemma

Research on moral dilemmas can be traced back to the famous philosophical problem of the “trolley dilemma” posed by Philippa Foot in 1967 [4,5,6], and later extended to the “flyover dilemma” in which participants must directly participate [7]. A survey on the “trolley dilemma” showed that 89% of respondents were willing to sacrifice one person to save the lives of five, but in the flyover dilemma only 12% agreed to push out the fat man [8]. This also illustrates the complexity of the moral dilemma itself.

Moral dilemmas are an important challenge for AVs in determining whether they will actually make it to the road. For AVs, the question is whether they should continue to drive as usual and hit most people on the road or suddenly swerve and hit only one person. Or does an autonomous vehicle need to act differently when the passenger’s life is at stake, rather than a pedestrian outside the vehicle [9]? The major autonomous driving companies believe that advanced driver assistance systems will continue to evolve, eventually leading to a future of zero autonomous driving accidents. However, they also acknowledge that at this stage all AVs may face the rare traffic situation of an unavoidable collision (moral dilemma) [10]. So, this kind of dilemma still exists.

To solve the dilemma, autonomous driving companies such as Baidu, Tesla, and Google have mainly targeted enhancements in the decision-making framework and technology of AVs. Many other companies have conducted research on moral dilemmas [10], as shown in Table 1. As can be seen above, current autonomous driving technologies are not sufficient to address moral dilemmas, and can only provide speed adjustment and remote manual supervision of vehicles. However, they are still actively exploring the ethical and social needs dimensions of autonomous driving decisions.

### 2.2. Deontologicalism and Utilitarianism

Moral dilemmas arise as a direct result of the diversity of moral norms. The ethics of autonomous driving focuses on the conflict between both utilitarianism and deontology. The current research on moral codes in resolving moral dilemma conflicts has taken four approaches: classical dilemma paradigm [11], process dissociation paradigm [12], CNI model [13], and CAN algorithm [14,15]. These methods can obtain the utilitarian or moral preferences of decision-makers under different conditions of influence.

Deontology asserts that the justification of an act comes from whether the act itself meets ethical standards, i.e., the AV must comply with certain basic norms. Related studies have mainly focused on rules and principles approaches in simple traffic environments; for instance, Thornton et al. [16] proposed the Three Laws of AVs, which specify the priority collision order of pedestrians, cars, and other objects in collision decisions. And Pagnucco et al. [17] presented a knowledge-based cognitive contextual algorithm for behavioral ethical reasoning. However, in the face of complex real-world traffic situations, deontology is difficult to design with full rule coverage.

By contrast, utilitarianism advocates the greatest happiness for the majority with the least damage, i.e., AVs should minimize the total harm caused by accidents. Most of the current decision-making research has chosen the utilitarianism of collision loss minimization. Researchers can use algorithms such as artificial intelligence to predict the severity of road traffic accidents based on state information such as the mass ratio, relative speed, and angle of the vehicle to the collision object for utilitarian loss calculations [18,19]. However, the disadvantages of utilitarianism are also obvious. If the decision algorithm does not protect the interests of vehicle owners, it will reduce the consumer subject’s desire to purchase or even cause them to refuse to purchase AVs.

Both deontology and utilitarianism have advantages and disadvantages in moral dilemma decision-making for AVs. Therefore, not only considering the deontological rulemaking but also the utilitarianism of minimizing losses in decision-making, and the relationship between the two needs to be properly coordinated.

### 2.3. Public and Individual Needs

In addition to ethics, it is important to address the contradiction between public and individual needs in automated vehicle decisions. Current studies have shown that factors such as age, gender, education, national culture, anxiety, vehicle safety, and traffic accident rates influence AVs decision-making [20,21]. In order to obtain public ideas, Edmond et al. [22] designed an online moral machine experiment platform that collected 40 million decision outcomes from millions of people in 233 countries and regions. The study showed that most people prefer to protect human beings, protect most lives, and protect young people, even though these choices may result in the death of passengers [22,23,24,25].

However, contrary to the results of moral machine studies, Bonnefon et al. [23] found that most people still prefer to purchase AVs that prioritize the safety of occupants in the vehicle. Moreover, Etienne et al. [26] argued that moral machines do not represent the general view of society. AVs need to be truly morally responsible and able to properly measure the risk to passengers and pedestrians as well as the safety of the vehicle [27]. These side effects reflect the uncertainty and vulnerability of public trust in automation [28].

In order to investigate the need for personalization, Gogol et al. [29] suggested allowing users to choose their own personal moral settings, and Contissa et al. [30] proposed setting their moral preferences by a continuously rotatable knob that turns left for “altruism” and right for “self-interest”. Moral decision-making principles cannot be determined by a single principle. Instead, personalized design without any restrictions and based solely on individual user preferences is likely to lead to a prisoner’s dilemma. Users will choose a suboptimal outcome that negatively affects society. Furthermore, human values [31], public moral preferences, and personal requirements need to be reflected in moral dilemma decisions for AVs.

In summary, the moral dilemma decision of AVs entails not only technical aspects but also involves moral ethical and social aspects. This study aimed to establish common principles that can be generally accepted by society and the public from the perspective of alleviating the conflict between utilitarianism and deontology and the contradiction between public and individual needs.

## 3. Materials and Methods

In order to explore common principles under the moral dilemma of AVs, this paper first selected a typical moral dilemma scenario, and then built the corresponding scenario based on five variables chosen, which included the number of sacrifices, vehicle passenger status, presence of children, decision-making control, and the law. In this section, we found relevant open-source data from Bonnefon et al. [23] for the selected moral dilemma scenarios and conducted a demographic analysis of the data. Finally, according to the needs of this paper and the characteristics of the open-source data, we chose the methods of gray correlation analysis, independent sample *t*-test, and analysis of covariance.

### 3.1. Scene Description

The classical moral dilemma scenario was selected as the basis of this paper. In order to explore the factors that influence the common principles of AVs confronting a moral dilemma, the design of this study contained five progressive questionnaire moral dilemma scenarios. The specific information of the scenarios and the decision-making choices of AVs are as follows:

1.The scenario with the number of lives sacrificed as the variable is that you are driving an AV on a main road with a speed limit, and 1/2/5/10/20/100 pedestrians suddenly appear on the road ahead. At this point, the AV has two choices: A. Stay on course and you will not be harmed, but will hit and kill the pedestrians suddenly appearing in front of you; B. Swerve suddenly to protect the safety of the pedestrians suddenly appearing in front of you, but the AV will crash into an obstacle and kill you as a passenger;2.The scenario with the passenger relationship as the variable is that you/you and your colleague/you and a family member are riding in an AV on a main road with a speed limit, and suddenly there are 10/20 pedestrians on the road ahead. At this point, the AV has two choices: A. Stay on course and you will not be harmed, but will hit and kill the pedestrians suddenly appearing in front of you; B. Swerve suddenly to protect the safety of the pedestrians suddenly appearing in front of you, but the AV will crash into an obstacle and kill you as a passenger;3.The scenario with the presence of children as a variable is that you/you and your family member/you and your children are riding in an AV on a main road with a speed limit, and 10/20 pedestrians suddenly appear on the road ahead. At this point, the AV has two choices: A. Stay on course and you will not be harmed, but will hit and kill the pedestrians suddenly appearing in front of you; B. Swerve suddenly to protect the safety of the pedestrians suddenly appearing in front of you, but the AV will crash into an obstacle and kill you as a passenger;4.The scenario in which the controlling subject of the decision is a human or a programmed computer variable involves you/others in an AV driving on a main road at the speed limit and 1/10 pedestrians suddenly appear on the road ahead. At this point, the AV has only two choices: A. Stay on course and you will not be harmed, but will hit and kill the pedestrians suddenly appearing in front of you; B. Swerve suddenly to protect the safety of the pedestrians suddenly appearing in front of you, but the AV will crash into an obstacle and kill you as a passenger;5.The scenario comparing the presence of illegal pedestrians and law-abiding pedestrians is that you are riding in an AV at high speed on a main road, and 1/10 pedestrians suddenly appear on the road ahead. The AV designer is programmed to offer three choices: A. Stay on course and you will not be harmed yourself, but will hit and kill the pedestrian who suddenly appears in front of you; B. Swerve suddenly to protect the pedestrian who suddenly appears in front of you, but the AV will crash into an obstacle and kill you as a passenger, or the AV will crash into the pedestrian on the side of the road and you will not be harmed yourself; C. Random choice: the car is programmed to randomly choose to stay on course or swerve.

The above five scenarios containing the number of sacrifices, vehicle passenger status, presence of children, decision-making control, and the law are summarized in Table 2, and Figure 1 shows a schematic diagram of the scenarios. In this study, it is assumed that all persons in the scenarios are adult males unless explicitly stated otherwise, and the number of people set in scenario 1 exceeds the actual road conditions just to explore common principles.

### 3.2. Questionnaire

This study used multi-scene, multi-perspective questionnaires. Participants would first be randomly assigned to one of five scenarios. Then, they would be tested by being randomly assigned to different situations in that scenario. Finally, the questionnaires were randomly selected from the passenger’s perspective, the pedestrian’s perspective, and the third party’s perspective.

Based on the need for data and judgment of the actual situation in this paper, it was difficult to obtain a large amount of questionnaire data. So, this study cited relevant open-source data from Bonnefon et al. [23], which were obtained from participants’ online questionnaires and included age, gender, religion, moral evaluation, purchase desire, fear value, and other relevant research parameters.

In order to conduct sample size and frequency statistics on the data, this paper used SPSS to perform a descriptive statistical analysis of the participant information. Table 3 shows the results of the demographic analysis of the participants in the five studies. Table 3 provides statistics on the number of participants in each study, analyzing the number, percentage, mean, and standard deviation of gender and age composition. The overall male-to-female ratio in this paper was similar, predominantly male (50.8%), and the age was mainly concentrated between 29 and 50 years old.

### 3.3. Gray Correlation Analysis

Gray correlation analysis is a mathematical and statistical analysis method that uses gray correlation degrees to describe the strength, magnitude, and order of the relationship between factors [32]. In this paper, there were many factors influencing decision-making in moral dilemmas, and the size and physical significance of the influencing factors indicators were different, which met the data requirements of gray correlation analysis. At the same time, in order to avoid interference in the results due to the overlapping information caused by multivariate covariance, this study finally chose gray correlation analysis to analyze and rank the factors influencing driving decisions in ethical dilemmas. The model construction processes are as follows.

#### 3.3.1. Data Processing

First, to determine the mapping values, $X_{0}^{\prime }=\left[X_{0}^{\prime }\left(1\right),X_{0}^{\prime }\left(2\right),\ldots ,X_{0}^{\prime }\left(n\right)\right]$ is assumed to be the reference data column and $X_{j}^{\prime }=\left[X_{j}^{\prime }\left(1\right),X_{j}^{\prime }\left(2\right),\ldots ,X_{j}^{\prime }\left(n\right)\right],\;\left(j=1,2,\ldots ,m\right)$ is the comparison data column.

Since the original data is dimensionless, it is assumed that the dimensionless reference data is listed as $X_{0}=\left[X_{0}\left(1\right),\ldots ,X_{0}\left(n\right)\right]$ and the dimensionless comparison data is listed as $X_{j}=\left[X_{j}\left(1\right),\ldots ,X_{j}\left(n\right)\right]$. The calculation formulas are shown in Equations (1) and (2).
(1)$$X_{0}^{}\left(k\right)=\frac{X_{0}^{\prime }\left(k\right)}{X_{0}^{\prime }},\;k=1,2,\ldots ,n.$$
(2)$$X_{j}^{}\left(k\right)=\frac{X_{j}^{\prime }\left(k\right)}{X_{j}^{\prime }\left(1\right)},j=1,2,\ldots ,m.$$

#### 3.3.2. Gray Correlation Calculation

We use ${△}_{jk}=\left|x_{0}\left(k\right)-x_{j}\left(k\right)\right|$ to represent the absolute difference between the reference data column and the comparison data column at point k, where △ indicates the total absolute difference, and the formula of △ is shown in Equation (3).
(3)$$△=\frac{\left[\sum_{j=1}^{m}\sum_{k=1}^{n}\left|x_{0}\left(k\right)-x_{j}\left(k\right)\right|\right]}{\left(m\times n\right)}$$

The formula for calculating the maximum and minimum values of ${△}_{jk}$ are shown in Equations (4) and (5) below:(4)$${△}_{min}=min_{j}min_{k}\left|x_{0}\left(k\right)-x_{j}\left(k\right)\right|$$
(5)$${△}_{max}=max_{j}max_{k}\left|x_{0}\left(k\right)-x_{j}\left(k\right)\right|$$

Then the gray correlation coefficients of x and y at k point are shown in Equation (6):(6)$$\xi_{jk}=\frac{△min+\rho △max}{\left|x_{0}\left(k\right)-x_{j}\left(k\right)\right|+\rho △max}$$

ρ is the discrimination coefficient, ρ∈(0,1); and the smaller the value of ρ, the greater the resolving ability. We usually picked ρ as 0.5.

The mean values of the correlation coefficients of each indicator and the corresponding element of the reference data column are calculated separately for each comparison data column to reflect the correlation between each comparison data column and the reference data column, which is called the correlation degree and expressed as $r_{j}$. The calculation formula is shown in Equation (7).
(7)$$r_{j}=\frac{1}{n}\overset{n}{\underset{k=1}{\sum }}\xi_{jk}$$

### 3.4. Statistics and Analysis

The data obtained from the questionnaires were statistically and analytically analyzed to investigate the influence of participants’ individual factors on psychological factors in moral dilemma decision-making. An independent samples *t*-test was chosen for the analysis of the effect of gender on psychological factors. This method was specifically used when the dependent variable was a fixed distance variable, and the independent variable was two mutually independent groups, and compared whether there was a significant difference between the means of the two groups. In this study, age was divided into 3 groups. In order to investigate the influence of different age groups on psychological factors, analysis of covariance (ANOVA) was chosen in this paper. This method is mainly applied to multiple groups of independent variables and analyzes the relationship between the dependent variable Y and the covariate X in each group. In summary, this paper used SPSS statistical software (version 24.0) for statistical analysis using the Pearson correlation, the independent samples *t*-test, and ANOVA.

## 4. Results

### 4.1. Results of Grey Correlation Analysis

Based on the established gray correlation analysis model, this study chose to use the decision results as the reference data column and the factors influencing the participants’ decisions as the comparison data column; among them, $Y_{1}$,$Y_{2}$ were used as individual factors, and $Y_{3}$,$Y_{4}$,$Y_{5}$ were used as psychological factors.

In this study, the decision outcome was chosen as the reference data column in building the gray correlation analysis model. The factors influencing participants’ decisions were used as the comparison data columns, where $Y_{1},Y_{2}$ were used as the individual factor, and $Y_{3},Y_{4},Y_{5}$ as the psychological factor. The quantitative information content of the data columns is shown in the following Table 4.

Subsequently, we preprocessed the questionnaire data. Since study 2 was similar to study 3, study 2 was chosen as the representative. The amount of data processed in this study was large, so only the preprocessed data results of study 1 are shown in Table 5 in the paper, and the preprocessed data results of other scenarios are shown in Table A1, Table A2 and Table A3.

According to the calculation process of the correlation degree, the results of the gray correlation degree under different scenarios obtained by using MATLAB for programming are shown in Table 6. It can be seen that the factors affecting the decision in different scenarios are different, so individual and psychological factors have little impact on decision-making.

### 4.2. t-Test and ANOVA Result

We further explored the influence of individual factors of participants on psychological factors and analyzed whether they have an impact on the setting of common principles of AVs. The results of the independent samples *t*-test and analysis of covariance processing using SPSS in this paper are shown in Table 7 and Table 8.

The results indicated that gender showed significance (*p* < 0.05) in terms of purchase, fear, and excitement values, with men having higher desire and expectation values than women. There were also significant differences in purchase intentions and expectations by age: the younger the participant, the higher the acceptance of AVs. In particular, the fear values in study 4 only differed significantly between age groups. This suggested that public fear values increase when innocent people are sacrificed, and that fear values increase with age.

In summary, psychological factors and individual differences had little influence on the design of common principles for decision-making in moral dilemmas for AVs, but there was a mutual influence, so that common principles need to be extracted from the common choices of participants for research.

## 5. Discussion

In this section, inductive cluster analyses of participants’ decision choices in each of the five moral dilemma scenarios are conducted in order to obtain common principles for AV decision-making. The moral dilemma scenario study relationships are as follows: study 1 varies the number of people sacrificed; study 2 varies the nature of the relationship between the occupants; study 3 further explores the identity and age of the occupants; study 4 explores the participants’ acceptance of car programming and human decision-making; study 5 analyzes the participants’ acceptance of three decisions designed for car programming and explores whether law-abiding people are protected.

### 5.1. Study 1: Number of Sacrifices

In study 1, participants were randomly assigned to five scenarios for decision selection based on the principle of multiple scenarios, including 1V1, 1V2, 1V5, 1V20, and 1V100. Figure 2 shows the number of all participants in study 1 who chose to go straight or swerve, and it could be seen that when there was only one sudden pedestrian intruder, more than half (77%) chose to stay and protect the passenger themselves. When there were two pedestrians and more, the number of people choosing to swerve at the expense of the passenger increased gradually as the number of pedestrians grew.

Figure 3. shows the participants’ preferred decision choices of future AVs in different situations. It can be seen that the participants’ choice of preferred decision for future AVs in case of moral dilemmas roughly matched the actual participation situation. As the number of pedestrians suddenly intruding increased, participants’ preference for AVs’ choices shifted more toward sacrificing themselves to protect the majority of pedestrians. Compared to Figure 2, the difference between straight ahead and swerving options decreases, indicating that people would make more cautious choices at the expense of passengers.

Combined with the overall analysis of Figure 2 and Figure 3, the common principle changes dynamically according to the number of people sacrificed. When there is 1V1, protecting the lives of passengers is the common choice of most people; when 1V2, the choice of protecting passengers or the suddenly intruding pedestrians is close and needs to be personalized according to the moral preference of passengers; when 1VN and N > 2, the greater the number of intruders, the more people choose to sacrifice themselves. So, protecting the lives of a larger number of people is the common choice of the public for AVs decision-making.

### 5.2. Study 2: Passenger Status

Participants in study 2 were randomly assigned to two situations (as passengers by themselves or with family/colleagues). Figure 4 shows the results of participants choosing to keep straight or swerve in different situations, and it could be seen that the decision choice was influenced by the passenger’s identity. When the rider’s identity changed, the number of decision choices changed as well. Figure 5 shows the percentage of choices in each case. When 1V10 and the occupant was himself, 78% of people were willing to choose to sacrifice themselves to protect the lives of pedestrians. When 2V20 and the occupant was their colleague or family member, although the choice still ended up with sacrificing the passenger to protect the majority of pedestrians, the percentage of those who chose to keep straight kept increasing. And this indicated that the participants’ desire to protect was stronger when the passenger was closer to them.

The results of study 2 showed that participants chose to protect the lives of the majority from an ethical and moral standpoint, despite their internal desire to protect those associated with them. Thus, in scenarios where passengers are of different statuses and are all adults, protecting the majority of lives remains a common principle. In the design of AVs, if the premise of protecting the majority of lives is not violated, the relevant decisions can be considered in conjunction with the social relationships of the occupants, which can enhance the level of consumer trust. For example, if a person with a social relationship to the occupant bursts in front of the vehicle on a road at low speed, a steering may be considered to injure the occupant and avoid injury to the pedestrian.

### 5.3. Study 3: Passenger Age

Study 2 analyzed the influence of passenger status on decision-making, so study 3 added children to the passengers to explore the degree of influence on decision-making. In study 3, participants were randomly assigned according to their passenger status composition. A moral score was then assigned to the choice of swerving in an autonomous vehicle for that situation and to the government’s choice of mandatory swerving with minimal casualties, with 100 being strongly in favor of swerving and 0 being strongly against swerving. As seen in Figure 6, the moral score of AVs implementing swerving under the principle of protecting majority life by the government is generally lower than 50. And it indicated that people do not accept government-mandated swerving of AVs.

Therefore, the moral score and the government-mandated moral score results were both lowest when the passenger was a child. It can be indicated that when there is 1V10 and one is a child, protecting the majority is still the common principle for decision-making in the moral dilemma of AVs. However, when there is 1V2 and one is a child, the principle of protecting children is superior to the principle of protecting the majority.

### 5.4. Study 4: Decision Object

Study 4 focused on participants’ perceptions of whether the vehicle decision was made by the programming algorithm or by a human in the presence of different numbers of sudden pedestrian intruders, and the effect of whether the participant was in the vehicle on the decision outcome. Participants counted the AV decision choices in this scenario, and the results are shown in Figure 7. When there were 10 pedestrians, the vast majority supported swerving, regardless of whether it was a human decision or a programming algorithm. When 1V1, the majority choice was to keep straight, and a higher percentage chose to keep straight under the programming algorithm. The explanation for this choice was that the sudden intrusion of a pedestrian was illegal, and the passenger preferred to protect himself by obeying the law, and the programming algorithm kept the passenger straight without his direct involvement and with less guilt.

Figure 8 represents the participants’ moral ratings of the AV’s choice to swerve in each of the four scenarios, A, B, C, and D. It could be seen that while the participants’ moral ratings of the swerving decision were all above 50 overall, indicating that the protection of most lives is not affected by the object of the decision. For human decision-making compared to programming algorithms, people expected programming algorithms to be more cautious, reflecting the state of distrust of AVs. When passengers were by themselves, people were more inclined to sacrifice themselves, and thus protecting others and the lives of the majority was more important, and people’s altruistic attributes could be reflected.

Figure 9 shows the participants’ ratings of their acceptance of the government’s law to force swerving under the principle of minimal loss, with a low average score. In contrast to Figure 8, Figure 9 shows that despite the perception that swerving was more moral, the acceptability was still low when forced by law, despite the latter forcing AVs to swerve to protect most lives.

In summary, it was clear that there was little variability in outcomes between human and programmed decisions in study 4, but humans still had difficulty accepting computer programming as utilitarian. Therefore, the decision subject has no direct influence on the decision outcome, and when there is 1V1, the protection of the law-abiding person remains the first common principle.

### 5.5. Study 5: Law Abiding

The number of options for decision-making and the number of law-abiding passersby at the roadside were added in study 5. Participants would be randomly assigned to a different number of sudden intruders and the presence or absence of roadside law-abiding pedestrians. In Figure 10 and Figure 11, passenger and pedestrian represent whether the swerving sacrifice is a passenger or a roadside law-abiding pedestrian.

Figure 10 presents the situation when there was only one sudden pedestrian intruder, and the moral ratings of participants for different decisions when swerving would either sacrifice passengers or sacrifice law-abiding pedestrians on the roadside. It shows that when 1V1, the moral score of choosing to keep straight was the same as that of swerving among sacrificed passengers; some people chose to leave it completely to the car programming, and it was hard for people to judge the trade-off, with some preferring a random choice of sacrifice. When 1V1, the moral score of choosing to keep straight over swerving among sacrificed roadside law-abiding pedestrians was higher, indicating that people thought it was more moral to save the lives of law-abiding people when the number of lives was the same.

When 1V10, it can be seen from Figure 11 that despite facing 10 illegal sudden pedestrians, participants still considered it more moral to swerve to save the lives of ten pedestrians (M > 50). Therefore, when there are equal numbers of law-abiding and law-breaking people in the scenario, protecting the lives of law-abiding people can be the first common principle of the AV, while when there are more law-breaking people in the scenario, protecting the lives of most people is the first common principle and protecting the lives of law-abiding people is the second common principle.

## 6. Conclusions and Recommendations

The current moral dilemma of AVs faces two difficulties. First, while most people agree to protect the lives of the greater number of people, they say they cannot accept the trend of AVs when they are set up to be completely utilitarian. Second, people are even less accepting of mandatory regulation by government regulators or autonomous car programmers.

Therefore, in this paper, we analyze the influencing factors of AV decision-making in moral dilemmas to establish a common decision that can change dynamically with the scenario and is generally accepted by society. There is no consistency in the influence of people’s individual and psychological factors on decision-making, and the level of influence changes dynamically with different scenario variables, while there is a significant influence between individual and psychological factors, and people of different ages and genders have different fear and expectation values for AVs. The specific common principles obtained in this paper are as follows:

1.When there is 1V1, that is, 1 passenger or curbside law-abiding pedestrian on the side of the road vs. 1 pedestrian who bursts in and breaks the law, protecting the passenger or protecting the law-abiding person on the side of the road is the common principle;2.When there is 1V2 there are two situations: (1) 1 passenger or curbside law-abiding pedestrian vs. 2 sudden intruders, where the AV decision is set according to the moral preference of the law-abiding person; (2) 1 child vs. 2 sudden intruders, where child protection is the common principle;3.When there is 1VN and N > 2, that is, 1 passenger or curbside law-abiding pedestrian vs. N sudden intruding pedestrians, protecting the lives of the majority is the common principle.

There are limitations to the above findings and suggestions for further research are given in order to eliminate the heterogeneity of the study. Firstly, the experimental scenarios should be designed to more closely match realistic and complex traffic environments, and driving choices should not be limited to straight ahead versus swerving [33,34]. Secondly, in order to obtain the influence of cultural and social values on common principles in different countries and regions, additional experimental data collection is needed. Finally, it is suggested that future research should incorporate prospect theory to combine autonomous driving with individual psychological and behavioral attitudes to eventually form a complete moral dilemma decision-making mechanism [35,36].

## Figures and Tables

**Figure 1 behavsci-12-00344-f001:**
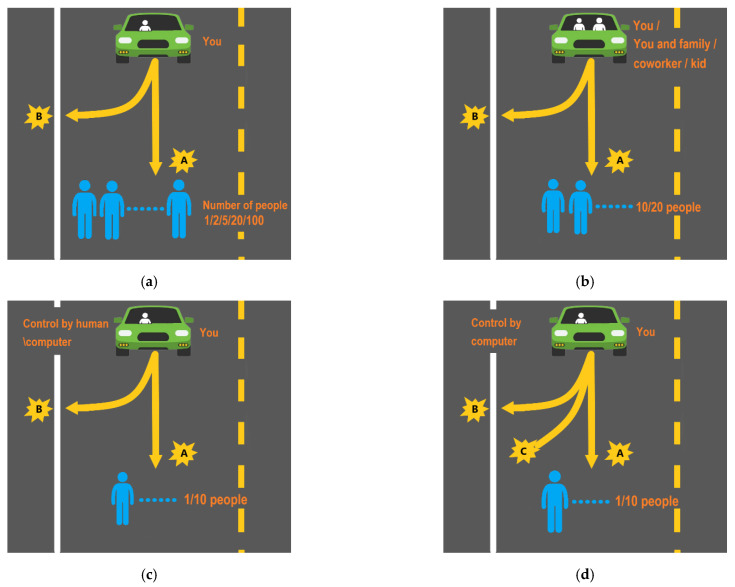
Scene diagram of studies. (**a**) study 1 scenario diagram; (**b**) study 2 and study 3 scenario diagram; (**c**) study 4 scenario diagram; (**d**) study 5 scenario diagram.

**Figure 2 behavsci-12-00344-f002:**
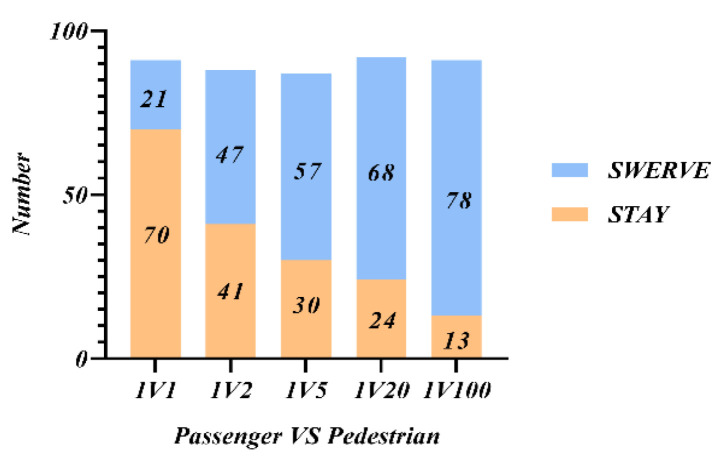
Participants’ preferred choice.

**Figure 3 behavsci-12-00344-f003:**
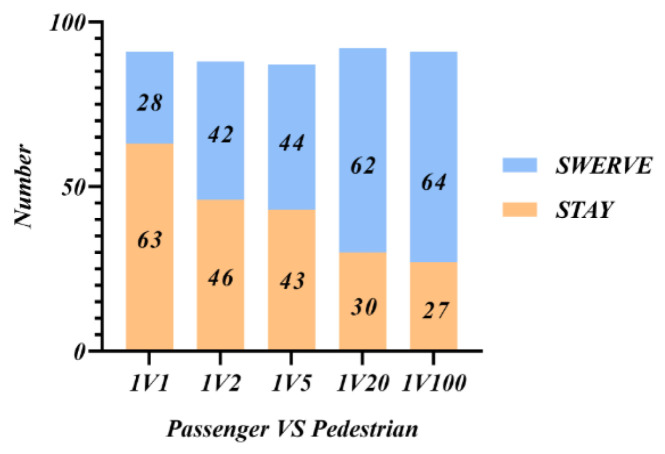
Participants’ future preference choice.

**Figure 4 behavsci-12-00344-f004:**
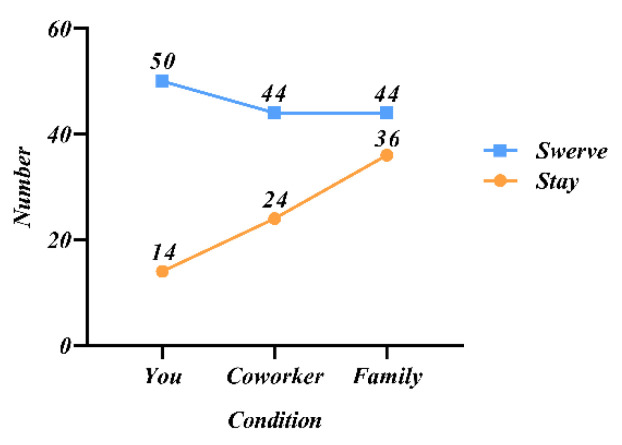
Participants’ preferred choice in different conditions.

**Figure 5 behavsci-12-00344-f005:**
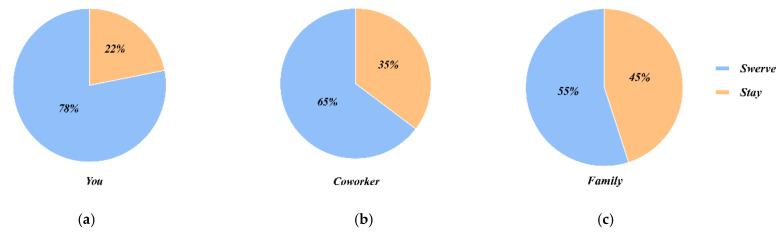
Participants’ choice of proportional sector at different passenger status. (**a**) you in the AV; (**b**) your coworker in the AV; (**c**) your family member in the AV.

**Figure 6 behavsci-12-00344-f006:**
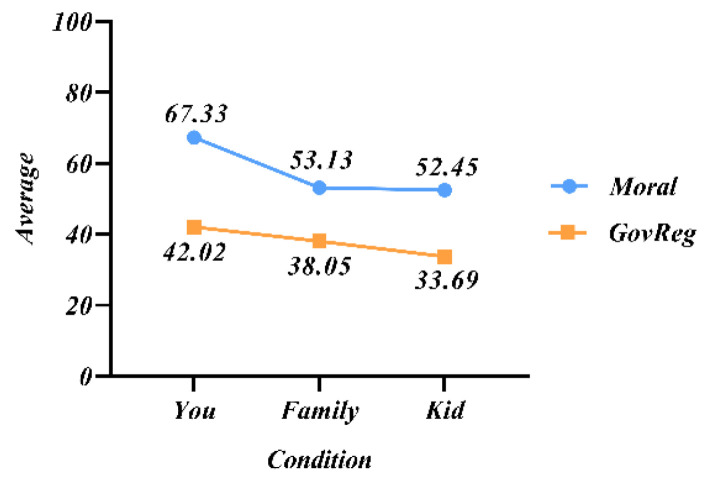
Mean values of participants’ moral evaluation scores on swerving choices with different passenger compositions.

**Figure 7 behavsci-12-00344-f007:**
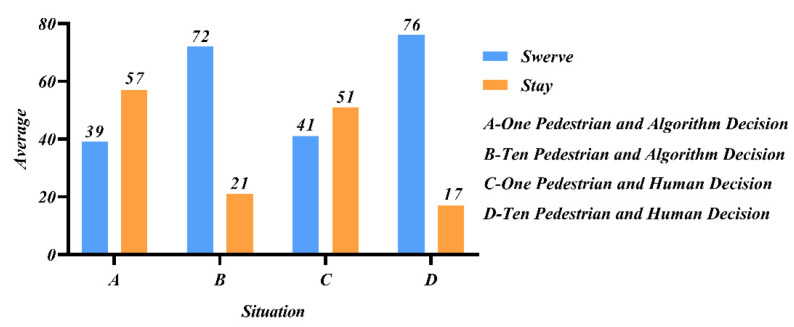
Participants’ preferences in different situations.

**Figure 8 behavsci-12-00344-f008:**
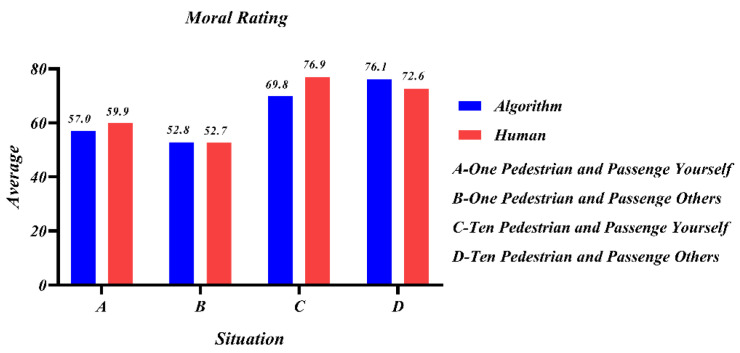
Mean of the moral evaluation scores for participants’ choice of swerving in different situations.

**Figure 9 behavsci-12-00344-f009:**
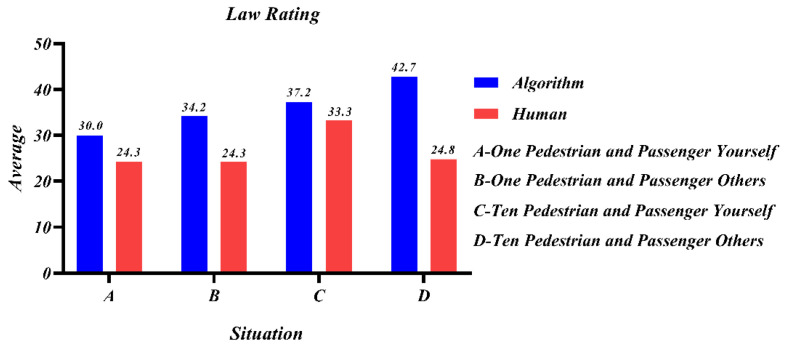
Mean of participants’ evaluation scores of legally enforced swerving in different situations.

**Figure 10 behavsci-12-00344-f010:**
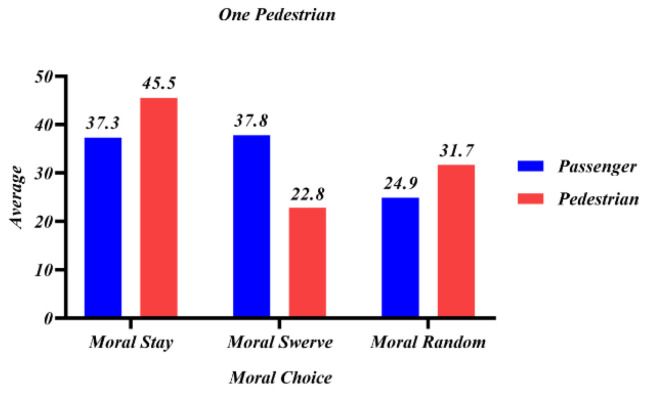
Mean values of participants’ evaluation scores for different moral choices when there was one sudden pedestrian intruder.

**Figure 11 behavsci-12-00344-f011:**
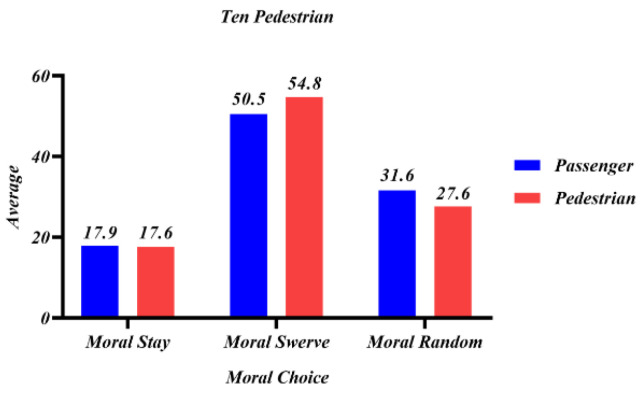
Mean values of participants’ evaluation scores for different moral choices when there were ten sudden pedestrian intruders.

**Table 1 behavsci-12-00344-t001:** Research advances in moral dilemmas for autonomous driving companies.

Autonomous Driving Company	Research Method
Intel	Reduce speed in advance when vehicle vision is obscured; choose to maintain autonomous control of the vehicle; use the Responsibility Sensitive Safety (RSS) model.
Mercedes-Benz and Bosch	Using Object and Event Detection and Response (OEDR) system to help autonomous driving systems handle traffic situations.
BMW	Set up a “black box” to store data that collates responsibility for accidents and is used to assign responsibility for people and machines in accidents.
Toyota	Establish clear rules in advance and set up a black box to store accident data for use in assigning responsibility in the event of a subsequent accident.
Uber	Manual supervision method, where the task specialist performs manual control of the vehicle in scenarios not included in the vehicle operation design field.
AutoX	Remote supervision method, where remote operators can check and correct the results of decisions.
Zoox	Remote supervision method, where a remote operator will remotely guide the vehicle in case of uncertainty.

**Table 2 behavsci-12-00344-t002:** Scene information of five studies.

Number	Passenger	Pedestrian Intruder	Law-Abiding Pedestrians	Decision Object	Decision Choice
Study 1	You	1/2/5/20/100	No	Human	A. Stay. Kill intruders.
					B. Swerve. Kill passenger.
Study 2	You/You and Coworker/Family Member	10/20	No	Human	A. Stay. Kill intruders.
					B. Swerve. Kill passenger.
Study 3	You/You and Family member/kid	10/20	No	Human	A. Stay. Kill intruders.
					B. Swerve. Kill passenger.
Study 4	You/Other people	1/10	No	Human/ Algorithm	A. Stay. Kill intruders.
					B. Swerve. Kill passenger.
Study 5	You	1/10	Yes	Algorithm	A. Stay. Kill intruders.
					B. Swerve. Kill Law-abiding pedestrians.
					C. Random. Choose A or B.

**Table 3 behavsci-12-00344-t003:** Demographical information for the studies.

Participant *	Items	Category	Frequency	Percent	M	SD
Study 1 449	Gender	Male	241	53.7%		
		Female	208	46.3%		
	Age	18–28	211	47.0%	23.94	2.78
		29–50	189	42.1%	36.28	6.08
		>50	49	10.9%	56.53	5.66
Study 2 212	Gender	Male	91	42.9%		
		Female	121	57.1%		
	Age	18–28	86	40.6%	24.48	2.33
		29–50	108	50.9%	35.78	6.34
		>50	18	8.5%	60.50	6.26
Study 3 391	Gender	Male	184	47.1%		
		Female	207	52.9%		
	Age	18–28	147	37.6%	24.06	2.61
		29–50	186	47.6%	37.16	6.26
		>50	58	14.8%	58.52	5.00
Study 4 374	Gender	Male	223	59.6%		
		Female	151	40.4%		
	Age	18–28	156	41.7%	24.54	2.59
		29–50	189	50.5%	36.19	5.78
		>50	29	7.8%	57.34	6.26
Study 5 267	Gender	Male	121	45.3%		
		Female	146	54.7%		
	Age	18–28	106	39.7%	23.81	2.90
		29–50	124	46.4%	36.94	6.01
		>50	37	13.9%	58.43	5.47

* The number of participants in five study.

**Table 4 behavsci-12-00344-t004:** Quantitative results of driving decision-making influencing factors.

	Mapping Name	Values
X	Decision-making: stay and swerve are marked as $X_{1}$ and $X_{2}$.	Give $X_{1}$ and $X_{2}$ an initial value respectively, using the percentage of the corresponding number of records in the total number of records.
$Y_{1}$	Gender: the number of men and women in the questionnaire scene are marked as $Y_{11}$ and $Y_{12}$.	Give $Y_{11}$ and $Y_{12}$ an initial value respectively, using the percentage of men and women participants in the total.
$Y_{2}$	Age: the number of participants in the questionnaire scene, 18–30 years old and over 30 years old are marked as $Y_{21}$ and $Y_{22}$.	Give $Y_{21}$ and $Y_{22}$ an initial value respectively, using the percentage of two age groups in the total.
$Y_{3}$	Fearful: the number of participants in the questionnaire scene, 1–3 scores and 4–7 scores are marked as $Y_{31}$ and $Y_{32}$.	Give $Y_{31}$ and $Y_{32}$ an initial value respectively, using the percentage of two- score groups in the total.
$Y_{4}$	Like to buy: the number of participants in the questionnaire scene,1–3 scores and 4–7 scores are marked as $Y_{41}$ and $Y_{42}$.	Give $Y_{41}$ and $Y_{42}$ an initial value respectively, using the percentage of two-score groups in the total.
$Y_{5}$	Excited: the number of participants in the questionnaire scene, 1–3 scores and 4–7 scores are marked as $Y_{51}$ and $Y_{52}$.	Give $Y_{51}$ and $Y_{52}$ an initial value respectively, using the percentage of two- score groups in the total.

**Table 5 behavsci-12-00344-t005:** Quantitative results of experimental data in study 1.

Number	*X*	$Y_{1}$	$Y_{2}$	$Y_{3}$	$Y_{4}$	$Y_{5}$
1	0.60	0.54	0.49	0.53	0.54	0.38
2	0.60	0.46	0.51	0.53	0.54	0.38
3	0.60	0.46	0.51	0.47	0.54	0.62
4	0.60	0.54	0.49	0.47	0.54	0.62
…	…	…	…	…	…	…
446	0.40	0.46	0.49	0.47	0.54	0.38
447	0.40	0.54	0.51	0.53	0.46	0.62
448	0.40	0.54	0.49	0.53	0.46	0.62
449	0.40	0.54	0.49	0.47	0.46	0.62

**Table 6 behavsci-12-00344-t006:** Grey relational degree.

	$Y_{1}$	$Y_{2}$	$Y_{3}$	$Y_{4}$	$Y_{5}$
Study 1	0.6142	0.6013	0.6024	0.6249	0.6815
Study 2	0.6100	0.5812	0.7025	0.7106	0.6343
Study 3	0.8156	0.8078	0.6044	0.6283	0.7736
Study 4	0.7226	0.6793	0.7413	0.6919	0.7439

**Table 7 behavsci-12-00344-t007:** Independent sample *t*-test for gender.

	Factor	Male	Female	T	*p*
Study 1	Buy	3.60 ± 2.09	3.04 ± 1.96	2.905	0.004 **
	Fearful	2.78 ± 1.84	3.70 ± 1.87	−5.216	<0.001 **
	Excited	4.55 ± 2.03	3.83 ± 2.11	3.667	<0.001 **
Study 2	Buy	3.22 ± 2.13	2.56 ± 1.71	2.420	0.017 *
	Fearful	3.04 ± 1.87	3.88 ± 1.73	−3.380	0.001 **
	Excited	4.07 ± 2.24	3.83 ± 1.94	0.815	0.416
Study 3	Enthusiasm	3.91 ± 1.69	2.99 ± 1.64	5.477	<0.001
Study 4	Buy	3.60 ± 2.01	2.81 ± 1.89	3.814	<0.001 **
	Fearful	2.58 ± 1.74	3.83 ± 1.84	−6.641	<0.001 **
	Excited	4.68 ± 1.95	3.53 ± 2.01	5.532	<0.001 **
Study 5	Buy	3.25 ± 2.14	2.38 ± 1.82	3.543	<0.001 **
	Fearful	4.34 ± 1.96	5.32 ± 1.64	−4.402	<0.001 **
	Excited	4.25 ± 2.13	3.23 ± 2.04	3.959	<0.001 **

* *p* < 0.05, ** *p* < 0.01.

**Table 8 behavsci-12-00344-t008:** ANOVA for age.

	Factor	18−28	29−50	>50	F	*p*
Study 1	Buy	3.67 ± 2.1	3.24 ± 1.98	2.29 ± 1.62	11.130	<0.001 **
	Fearful	3.07 ± 1.87	3.21 ± 1.93	3.75 ± 1.94	2.495	0.084
	Excited	4.62 ± 2.02	4.03 ± 2.13	3.18 ± 1.80	9.763	<0.001 **
Study 2	Buy	3.24 ± 2.14	2.58 ± 1.68	2.84 ± 1.92	3.210	0.042 *
	Fearful	3.37 ± 1.98	3.69 ± 1.67	3.48 ± 1.84	1.003	0.368
	Excited	4.40 ± 2.03	3.63 ± 2.02	3.50 ± 2.28	3.780	0.024 *
Study 3	Enthusiasm	3.78 ± 1.68	3.28 ± 1.72	3.00 ± 1.74	5.617	0.004
Study 4	Buy	3.51 ± 2.02	3.22 ± 1.98	2.41 ± 1.74	3.945	0.020 *
	Fearful	2.87 ± 1.91	3.08 ± 1.79	4.21 ± 2.01	6.336	0.002 **
	Excited	4.60 ± 1.99	4.06 ± 2.01	3.17 ± 2.21	7.286	0.001 **
Study 5	Buy	3.21 ± 2.13	2.62 ± 1.95	2.03 ± 1.59	5.541	0.004 **
	Fearful	4.72 ± 1.90	4.84 ± 1.86	5.47 ± 1.58	2.304	0.102
	Excited	4.11 ± 2.17	3.53 ± 2.10	3.03 ± 2.02	4.279	0.015 *

* *p* < 0.05, ** *p* < 0.01

## Data Availability

The data presented in this study are available in www.sciencemag.org/content/352/6293/1573/suppl/DC1 accessed on 3 August 2022.

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
