# Peer review of "A Study of Common Principles for Decision-Making in Moral Dilemmas for Autonomous Vehicles"

_behavsci, 2022, doi:10.3390/bs12090344_

Round 1

Reviewer 1 Report

1. In the introduction, you should show how currently autonomous vehicle companies address the problem (decision-making in moral dilemmas for Autonomous Vehicles ). What are the limitations? 

2. In the introduction, I recommend that authors state their contributions as bullets to be easy to understand by readers

3. You need a paragraph at the end of the introduction section to show how your paper is structured.

4. What are the main gaps in previous studies in this field? You need to improve your related work discussion. For example, A comparison table should be added describing, all the plus points of the proposed technique with the existing methods.

5. In figure 1 (a), do you mean “number of people” ? instead of “people number”

6. All abbreviations should be defined in full the first time they appear in the title, abstract, main text, and figure or table captions, even if they are well known in the field. E.g. SPSS

7. It is recommended to cite the following sentences:

a. “ That can lead to accidents and even life threatening accidents when faced with emergencies such as malfunctioning sensor devices or sudden encounters with pedestrians.”

b.  “ Gray correlation analysis is a mathematical and statistical analysis method that uses gray correlation degree to describe the strength, magnitude, and order of the relationship between factors.”

8. How do the authors select the number of pedestrians for each scenario? and why?

9. How do the authors validate their work?

10. Some of the challenges encountered during the study can be highlighted, and future recommendations can be added at the end of the conclusion. Retitle conclusion as conclusion and recommendation.

Author Response

Dear Reviewer,

We appreciate your warm work earnestly. Those comments are valuable and very helpful. We have read through comments carefully and have made corrections. The comments and corresponding responses and detailed revisions are included in the attachment titled " Response to Reviewer Comments". 

Once again, thank you very much for your comments and suggestions.

Best regards!

Yours sincerely,

Shufeng Wang

Author Response

(The authors gave the same response as above.)

Reviewer 3 Report

This paper analyzes the influencing factors of AV decision making under moral dilemma to establish a common decision that can change dynamically with the scenario and is generally accepted by the society. Some revisions and clarifications are needed to fully appreciate the manuscript.

It is suggested to divide the introduction into two parts: introduction and review. The current version puts too much review content in the introduction, which makes the readers cannot get the main point directly at the beginning. Focus on background and meaning as well as a brief summary about the contents in the introduction. The existing gaps could be given in the review section.

While I appreciate the vast literature reported, its discussion is not well organized. At the moment, it looks like an unstructured description of current papers, listed just one after the other. It needs to be reorganized around the specific objective of this paper with the aim of clarifying the choices made in this paper and why these are relevant.

Especially, the authors have missed important recent literature: “The Moral Machine experiment”.

In the surveys, there are no options like “I prefer not to choose”. The answer has to be one of two opposite options, which forces the respondent to select one that they may not really want to choose. The problem should be mentioned in the limitation section by referring to two relevant literature.

Di, Xuan, Henry X. Liu, Shanjiang Zhu, and David M. Levinson. "Indifference bands for boundedly rational route switching." Transportation 44, no. 5 (2017): 1169-1194.

Ortúzar, J. de D. (2021) ‘Future transportation: Sustainability, complexity and individualization of choices’

 The sample size for study 5 is not large. Please add some justifications about the validity of using these data for drawing the main conclusions in the paper.

In the discussion sections, there are large variations in the behaviors among different people. How do you consider this issue? Corresponding clarifications should be added. If the behavioral heterogeneity is not considered, at least a corresponding limitation statement should be provided to clarify this by adding the references.

The conclusion should deliver the main findings that are general rather than specific for the used scenarios. In the current version, the authors summarize the results of the experiments but not in an insightful way. It is suggested to include more general conclusions rather than results from the scenarios.

Limitation statements should be added in a more detailed way after conclusions as many aspects need to be improved. For instance, is the finding general? How can further research improve present outcomes? An interesting future study direction is to use some quantitative behavioral models to further investigate the topic based on the collected data, such as hybrid choice models and prospect theory (Gao et al. 2021; Li and Hensher et al. 2018). It is recommended that future study point is added in the discussions.

Z. Li, D. Hensher, Prospect theoretic contributions in understanding traveller behaviour: a review and some comments

Gao, K., Yang, Y. and Qu, X. (2021) Diverging effects of subjective prospect values of uncertain time and money

The writing should be checked and revised in terms of grammar, equations, and expressions.

Author Response

(The authors gave the same response as above.)

Round 2

Reviewer 1 Report

No comments thanks 

Reviewer 2 Report

Thank you for responding to my comments. There are still a few minor mistakes in the document that should be fixed by proofreading.

Reviewer 3 Report

Thank the authors for addressing my comments. There are still some minor typos in the manuscript, which should be corrected via a proofreading.